# Usefulness of Modified CEUS LI-RADS for the Diagnosis of Hepatocellular Carcinoma Using Sonazoid

**DOI:** 10.3390/diagnostics10100828

**Published:** 2020-10-15

**Authors:** Katsutoshi Sugimoto, Tatsuya Kakegawa, Hiroshi Takahashi, Yusuke Tomita, Masakazu Abe, Yu Yoshimasu, Hirohito Takeuchi, Yoshitaka Kasai, Takao Itoi

**Affiliations:** Department of Gastroenterology and Hepatology, Tokyo Medical University, Tokyo 160-0023, Japan; azusaktk36@gmail.com (T.K.); h.takahashi627@gmail.com (H.T.); toyutoyu6312@gmail.com (Y.T.); abechdesu@gmail.com (M.A.); yoshibo.you@gmail.com (Y.Y.); hirohito@yf6.so-net.ne.jp (H.T.); ykasai@tokyo-med.ac.jp (Y.K.); itoi@tokyo-med.ac.jp (T.I.)

**Keywords:** ultrasound, contrast media, Sonazoid, hepatocellular carcinoma, CEUS, LI-RADS

## Abstract

The Contrast-Enhanced Ultrasound Liver Imaging Reporting and Data System (CEUS LI-RADS) was introduced for classifying suspected hepatocellular carcinoma (HCC). However, it cannot be applied to Sonazoid. We assessed the diagnostic usefulness of a modified CEUS LI-RADS for HCC and non-HCC malignancies based on sensitivity, specificity, positive predictive value (PPV), and negative predictive value (NPV). Patients with chronic liver disease at risk for HCC were evaluated retrospectively. Nodules ≥1 cm with arterial phase hyperenhancement, no early washout (within 60 s), and contrast defects in the Kupffer phase were classified as LR-5. Nodules showing early washout, contrast defects in the Kupffer phase, and/or rim enhancement were classified as LR-M. A total of 104 nodules in 104 patients (median age: 70.0 years; interquartile range: 54.5–78.0 years; 74 men) were evaluated. The 48 (46.2%) LR-5 lesions included 45 HCCs, 2 high-flow hemangiomas, and 1 adrenal rest tumor. The PPV of LR-5 for HCC was 93.8% (95% confidence interval (CI): 82.8–98.7%). The 22 (21.2%) LR-M lesions included 16 non-HCC malignancies and 6 HCCs. The PPV of LR-M for non-HCC malignancies, including six intrahepatic cholangiocarcinomas, was 100% (95% CI: 69.8–100%). In conclusion, in the modified CEUS LI-RADS for Sonazoid, LR-5 and LR-M are good predictors of HCC and non-HCC malignancies, respectively.

## 1. Introduction

Contrast-Enhanced Ultrasound (CEUS) has been available since the late 1990s as a technique for characterizing liver nodules. With the development of low mechanical index imaging and second-generation contrast agents, real-time assessment of lesion perfusion in the arterial, portal, and delayed phases has become possible, providing useful information for the characterization of liver nodules, especially hepatocellular carcinoma (HCC) [1,2].

For years, CEUS has been used in Europe and Asia as a first-line diagnostic modality for HCC, and it has been endorsed by several national and international professional societies [3,4]. However, the HCC guidelines of the American Association for the Study of Liver Diseases [5] in the United States do not accept CEUS as a diagnostic technique for HCC because of the possibility that HCC may be misdiagnosed as intrahepatic cholangiocarcinoma (ICC) [6].

Considering this matter in greater detail, a number of studies have reported that the onset of washout from ICC is usually earlier (within 1 min) than that from HCC and that the degree of washout from ICC is greater than that from HCC [7,8]. When these characteristics are taken into account, CEUS provides high sensitivity and a high positive predictive value for the diagnosis of HCC.

To improve the diagnostic accuracy for HCC and to facilitate communication among radiologists and between radiologists and other physicians, the American College of Radiology has developed the Contrast-Enhanced Ultrasound Liver Imaging Reporting and Data System (CEUS LI-RADS) as a standardized reporting system for liver nodules in patients at risk for HCC [9].

Unfortunately, the current version of CEUS LI-RADS (version 2017) is applicable only to pure blood pool contrast agents such as Lumason (Bracco Diagnostics, Monroe Township, NJ, USA) and Definity (Lantheus Medical Imaging, Billerica, MA, USA), but not to combined blood pool and Kupffer cell contrast agents such as Sonazoid (GE Healthcare, Oslo, Norway). This is because pure blood pool agents provide effective arterial phase hyperenhancement and ensure pure contrast agent washout from malignant nodules.

Sonazoid is composed of microbubbles of perfluorobutane gas coated with hydrogenated egg phosphatidylserine sodium. It was approved for the imaging of focal liver lesions in 2007 in Japan, 2012 in Korea, 2017 in Taiwan, and 2018 in China. In Sonazoid CEUS examinations, the typical dynamic enhancement patterns of HCC are hyperenhancement in the arterial phase followed by iso-/hypo-enhancement in the portal phase and defects in the Kupffer phase (post-vascular phase), the same as for blood pool agents [10]. In more than 97% of cases, this pattern corresponds to HCC [11]. Thus, the diagnostic accuracy of Sonazoid for HCC is comparable to that of pure blood pool agents.

We have developed a modified CEUS LI-RADS which is also applicable to Sonazoid. The aim of the present study was to evaluate the diagnostic performance of this modified CEUS LI-RADS in patients at risk for HCC.

## 2. Materials and Methods

This study was reviewed and approved by Tokyo Medical University ethics review board (T2019-0179, 24 January 2020). The requirement to obtain written informed consent was waived for this retrospective study.

### 2.1. Patients

A clinical/pathological database was used to retrospectively identify 430 consecutive patients with risk factors for HCC who presented with untreated liver nodules and who underwent CEUS at our institution between March 2017 and April 2020.

The inclusion criteria were as follows: (a) age 20 years or older, (b) visible liver nodule in baseline ultrasound (US), (c) availability of a CEUS examination that conformed with our CEUS protocol and included vascular phase and Kupffer phase information (as described below), and (d) availability of an accepted diagnostic reference standard (as described below).

The exclusion criteria were as follows: (a) local relapse of a previously treated lesion, (b) cirrhosis due to a vascular disorder such as Budd–Chiari syndrome or cardiac congestion, (c) diffuse HCC, and (d) poor image quality for any reason.

### 2.2. Ultrasound (US) Examination

Conventional gray-scale and CEUS examinations were performed using an Aplio *i*800 diagnostic ultrasound system (Canon Medical Systems, Tochigi, Japan) equipped with a 3.5-MHz convex transducer (PVT-475BT; Canon Medical Systems). The number, size, location, and echogenicity of lesions were assessed. Pulse inversion harmonic imaging was used for CEUS examinations with a low mechanical index of 0.1–0.2 and a dynamic range of 45 dB. The second-generation US contrast agent Sonazoid (GE Healthcare) was injected as a 0.5-mL bolus into an antecubital vein via a 21-gauge peripheral intravenous cannula, followed by a 10-mL saline flush. A timer was started at the time of contrast agent injection. Images were recorded continuously as a cine clip for a period of 60 s immediately after injection of the contrast agent (for evaluation of the vascular phase), after which the scan was frozen. After a waiting period of approximately 10 min from the time of contrast agent injection to permit pooling of the agent in the liver parenchyma, enhancement of the lesion was observed using a sweep scan and images were recorded (for evaluation of the Kupffer phase or post-vascular phase). The sequence that was followed in the CEUS protocol is shown in Figure 1. The CEUS images were transferred as a set to an image workstation (Vitrea; Canon Medical Systems) for later evaluation.

### 2.3. Reference Standard

All malignant lesions, including both HCC and non-HCC malignancies, were diagnosed based on the findings of histopathologic examination. The reference standard for benign lesions was either histopathologic assessment or typical imaging features on dynamic CT or MRI with no change in size over a minimum 1-year follow-up period. The criteria of the modified CEUS LI-RADS are shown in Table 1.

### 2.4. Contrast-Enhanced Ultrasound (CEUS) Image Assessment

One hepatologist with more than 15 years of experience in liver CEUS, who was blinded to the reference standard results and other imaging findings for the liver nodules, independently reviewed the CEUS examinations at the dedicated workstation (Vitrea) and assigned each nodule to a category according to the modified CEUS LI-RADS, which is based on CEUS LI-RADS (2017 version) [9]. The criteria are shown in Table 1. Briefly, the main difference between CEUS LI-RADS (2017 version) and the modified CEUS LI-RADS is that the former includes portal phase and late phase washout as major imaging features, while the latter includes Kupffer phase findings as a major imaging feature. The other criteria in the modified CEUS LI-RADS are almost the same as those in CEUS LI-RADS (2017 version). In addition, to investigate the reasons that HCC nodules may show different patterns of enhancement, the correlations between the degree of histopathologic HCC differentiation and the modified CEUS LI-RADS category were assessed.

### 2.5. Statistical Analysis

Qualitative data are presented as numbers and percentages, and quantitative data are presented as median and interquartile range (IQR). The overall diagnostic capabilities of the modified CEUS LI-RADS were assessed in terms of diagnostic accuracy, sensitivity, positive predictive value (PPV), and negative predictive value (NPV) with the 95% confidence interval (CI). Categorical variables were compared using the paired chi-square test. All statistical analyses were performed using software (JMP version 14; SAS, Tokyo, Japan). Values of *p* < 0.05 were considered as statistically significant.

## 3. Results

### 3.1. Patient Characteristics

Based on the selection criteria, a total of 104 nodules in 104 patients were included in the study. The clinical characteristics of the patients, including age, sex, liver disease etiology, nodule size, and tumor histopathologic findings, are shown in Table 2. The median age of the 104 patients was 70.0 (IQR: 54.5–78.0) years, and 74 (71.2%) of the patients were men. Among the 91 patients who undertook liver histological examinations, 80 patients (87.9%) had liver cirrhosis. The median size of the liver nodules was 17.9 (IQR: 13.1–28.2) mm. Based on the reference standard, 91 lesions were diagnosed by histopathologic examination and 13 were diagnosed by contrast-enhanced CT or contrast-enhanced MRI with minimum 1-year follow-up.

### 3.2. Distribution of Nodules in the Modified CEUS LI-RADS Categories

The percentages of HCC and non-HCC malignancies in each category and distribution of nodules in the modified CEUS LI-RADS categories are shown in Table 3 and Table 4. There were 48 nodules (46.2% of the total) in the LR-5 category, of which 45 (93.8%) were HCC. There were 22 nodules (21.2%) in the LR-M category, of which 15 (68.2%) were non-HCC malignancies. There were no malignant nodules in the LR-1 and LR-2 categories. There were 10 nodules (9.6%) in the LR-3 category, of which 7 (70.0%) were HCC. These HCCs displayed a CEUS pattern of iso-iso (4/7), iso-hypo (2/7; both of which were less than 1 cm in diameter), and hypo-iso (1/7). There were 16 nodules (15.3%) in the LR-4 category, of which 6 (37.5%) were HCC. These HCCs displayed a CEUS pattern of hyper-iso (15/16) and hyper-hyper (1/16). Six LR-4 lesions were focal nodular hyperplasia (FNH), all of which displayed the hyper-iso pattern. Two LR-4 lesions were hemangiomas, all of which displayed the hyper-iso pattern, consistent with high flow hemangiomas.

### 3.3. Imaging Characteristics of All Liver Nodules

The US characteristics of the 104 liver nodules studied are shown in Table 5. Arterial phase hyperenhancement was observed in 81 nodules, including 77 with diffuse hyperenhancement, 3 with peripheral nodular enhancement, and 1 with rim enhancement. Early washout starting within 60 s was observed in 22 nodules, of which 6 (27.3%) were demonstrated to be HCC and 16 (72.7%) were demonstrated to be non-HCC malignancies. In the Kupffer phase, 73 nodules were hypoechoic. Of these 73 nodules, 53 (72.6%) were demonstrated to be HCC and 16 (30.2%) were demonstrated to be non-HCC malignancies.

### 3.4. Diagnostic Performance of the Modified CEUS LI-RADS LR-5 and LR-M Categories

The diagnostic performance of the modified CEUS LI-RADS LR-5 and LR-M categories is shown in Table 6. The sensitivity, specificity, PPV, NPV, and diagnostic accuracy of LR-5 for HCC were 70.3% (95% CI: 57.6–81.1%), 92.5% (95% CI: 79.6–98.4%), 83.8% (95% CI: 82.8–98.7%), 66.1% (95% CI: 52.2–78.2%), and 78.7% (95% CI: 69.7–86.2%), respectively. The sensitivity, specificity, PPV, NPV, and diagnostic accuracy of LR-M for non-HCC malignancies were 68.2% (95% CI: 45.1–86.1%), 100% (95% CI: 93.5–100%), 100% (95% CI: 69.8–100%), 92.1% (95% CI: 84.5–96.8%), and 93.3% (95% CI: 86.6–97.3%), respectively.

### 3.5. Modified CEUS LI-RADS Categories and Degree of Histopathologic Differentiation of HCC

The histopathologically proven HCCs (*n* = 64) included 28 well-differentiated, 32 moderately differentiated, and 4 poorly differentiated HCCs. The relationship between the modified CEUS LI-RADS categories and the degree of HCC differentiation is shown in Table 7. Of the six HCCs classified as LR-M, four (66.7%) were poorly differentiated HCCs. In contrast, no poorly differentiated HCCs were classified as LR-5. However, there was no difference in the percentage of poorly differentiated HCCs between LR-M and LR-5.

## 4. Discussion

The American College of Radiology’s CEUS LI-RADS has been developed as a standardized reporting system for liver nodules in patients at risk for HCC [9]. Although the CEUS LI-RADS algorithm is only applicable to CEUS examinations performed using pure blood pool contrast agents such as Lumason and Definity, the combined bold pool and Kupffer cell contrast agent Sonazoid is also useful for the diagnosis of hepatic nodules, including HCC [11]. Thus, the objective of our research was to develop an algorithm that includes Sonazoid with only minimal modification of the existing CEUS LI-RADS algorithm.

The results of the present study showed that the modified CEUS LI-RADS LR-5 category had a high PPV of 93.8% for HCC. This validated arterial phase hyperenhancement without early washout followed by defects in the Kupffer phase (post-vascular phase) as the diagnostic criteria for LR-5 nodules. The results also showed that the modified CEUS LI-RADS LR-M category had a PPV of 100% for non-HCC malignancies, including metastases and ICC. Moreover, one of the strengths of the present study was that the standard reference for all malignant lesions was based on histopathologic examination. These results are extremely important for clarifying the potential usefulness of CEUS for the diagnosis of HCC, demonstrating that the risk of misdiagnosing HCC as ICC (which is the main reason that CEUS is not included in the Western HCC diagnostic guidelines) can be largely avoided [6].

Among the sixty-four HCCs in the present study, washout starting within 60 s was observed in six nodules (9.4%). These nodules were therefore classified as LR-M. Moreover, four (66.7%) of these nodules were poorly differentiated tumors, which is in line with the findings of previous studies [12,13]. However, this finding is actually rather advantageous, because poorly differentiated HCCs have higher malignant potential than other types of HCCs and are therefore more similar in nature to metastases [14]. For example, one of the poorly differentiated HCCs in our study was treated by radiofrequency ablation, but aggressive local tumor progression was observed after treatment (Figure 2). Thus, the finding of early washout (<60 s) may be a relative contraindication to local ablation therapy.

This study included ten hemangiomas, of which two were miscategorized as LR-5. These were all high-flow-type hemangiomas which showed hyperenhancement in the arterial phase and hypo-enhancement in the Kupffer phase [15]. If we had used portal phase information, these lesions would not have been categorized as LR-5, but as LR-4. However, this finding is not particularly worrisome, because it is relatively infrequent and, more importantly, because diffusion-weighted imaging is extremely accurate in establishing a diagnosis of hemangioma (Figure 3) [16]. Thus, such CEUS misdiagnosis of probable HCC would usually become clear at the time of panoramic imaging, which should always be performed for staging before conducting any treatment.

This study also included 6 ICCs and 10 metastases. All were classified as LR-M with early washout and clear contrast defects in the Kupffer phase. Thus, the PPV for non-HCC malignancies was 100%. Sonazoid may therefore be more suitable for the diagnosis of non-HCC malignances. In addition, in the Kupffer phase, parenchymal enhancement is stable and persists for at least 2 h [17]. This means that Sonazoid may be suitable not only for the diagnostic assessment of non-HCC malignancies, but also for the initial detection of these malignancies [18].

In the CEUS LI-RADS LR-M criteria, arterial phase rim enhancement is one of the major imaging features, as well as early washout and marked washout. However, in our series, only 1 (ICC) of the 16 non-HCC malignancies showed rim enhancement (6.3%). CEUS can be used to observe lesions in real-time, and microbubbles have a high sensitivity for blood flow [19], so we have frequently encountered cases with discordance in image findings, showing rim enhancement in the arterial phase of dynamic CT and MRI, but uniform enhancement in the arterial phase of CEUS (Figure 4) [20]. Therefore, we should consider rim enhancement to be a finding with high specificity, but low sensitivity.

The present study had several notable limitations. First, we used retrospective data from a single-center experience involving patients with chronic liver disease, which might not translate to other centers. Future studies should include patients from multiple centers and employ different diagnostic ultrasound systems. Second, although all malignant lesions were ensured by histological examination, almost half of benign nodules (54.2%) was not. Finally, our proposed modified CEUS LI-RADS must be validated in future independent cohorts examined using Sonazoid prior to its adoption by LI-RADS.

In conclusion, the findings of the present study validated the diagnostic usefulness of our proposed modified CEUS LI-RADS algorithm for the combined blood pool and Kupffer cell contrast agent Sonazoid. The results showed that the modified CEUS LI-RADS LR-5 category is the optimal diagnostic tool for histopathologically proven HCC and that the modified CEUS LI-RADS LR-M category is the optimal diagnostic tool for histopathologically proven non-HCC malignancies (including ICC and metastases) with an excellent PPV. This modification to the CEUS LI-RADS algorithm to include Sonazoid should therefore be considered for incorporation into the current CEUS LI-RADS algorithm when it is revised in the future.

## Figures and Tables

**Figure 1 diagnostics-10-00828-f001:**
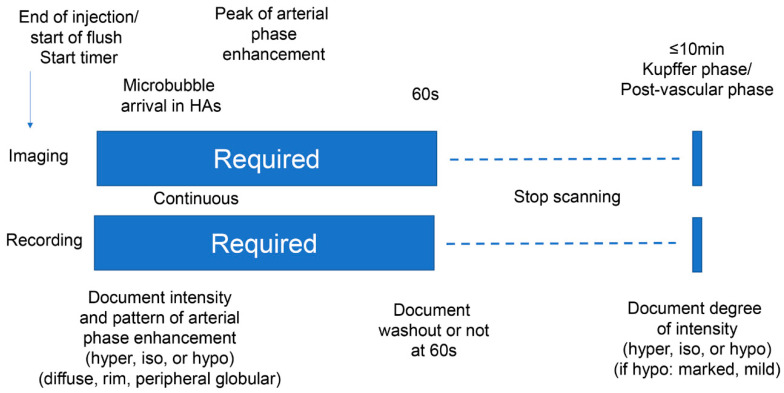
Sonazoid Contrast-Enhanced Ultrasound protocol.

**Figure 2 diagnostics-10-00828-f002:**
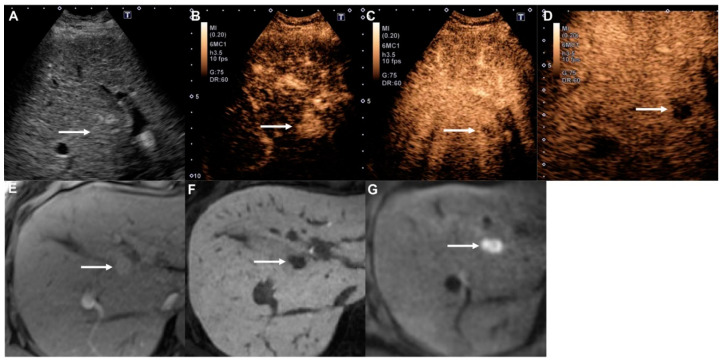
Poorly differentiated hepatocellular carcinoma in a 58-year-old woman with hepatitis B cirrhosis. (**A**) Gray-scale ultrasound shows a slightly hyperechoic mass (arrow) in the liver. (**B**) The nodule (arrow) shows hyperenhancement at 14 s in the arterial phase of Contrast-Enhanced Ultrasound (CEUS). (**C**) Early washout is apparent (arrow) at 40 s in the portal venous phase. (**D**) The nodule (arrow) shows a contrast defect in the Kupffer phase. (**E**) The arterial phase of Gd-EOB-DTPA-enhanced MRI (EOB-MRI) shows homogeneous hyperenhancement (arrow) in the liver. (**F**) The hepatobiliary phase of EOB-MRI shows a clear hypointense mass (arrow). (**G**) Diffusion-weighted imaging (*b* = 800 s/mm^2^) shows a clear hyperintense mass (arrow), indicating diffusion restriction.

**Figure 3 diagnostics-10-00828-f003:**
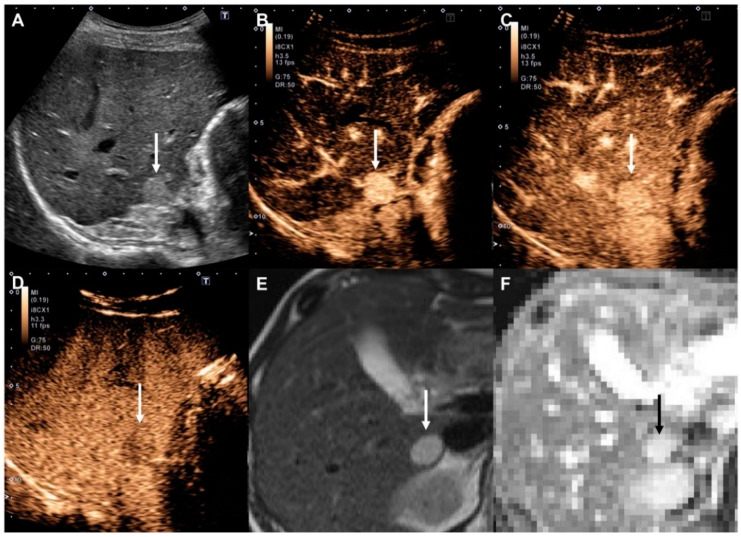
High-flow hemangioma in a 48-year-old woman with chronic hepatitis B. (**A**) Gray-scale ultrasound shows a clear hyperechoic mass (arrow) in the liver. (**B**) The nodule (arrow) shows hyperenhancement at 22 s in the arterial phase of Contrast-Enhanced Ultrasound (CEUS). (**C**) Enhancement persists at 51 s in the portal venous phase (arrow). (**D**) In the Kupffer phase, the nodule appears as a slightly hyperechoic mass (arrow) in the liver. (**E**) Unenhanced T2-weighted MRI shows a clear hyperintense mass (arrow) in the liver. (**F**) The apparent diffusion coefficient (ADC) map shows a hyperintense mass (black arrow), indicating lower diffusion restriction.

**Figure 4 diagnostics-10-00828-f004:**
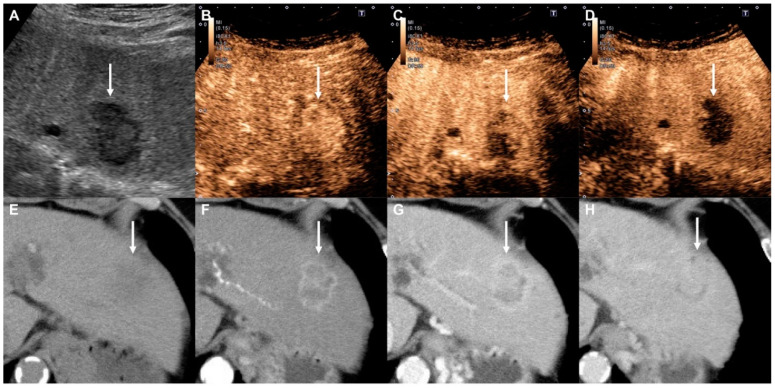
Intrahepatic cholangiocarcinoma in a 69-year-old woman with alcoholic cirrhosis. (**A**) Gray-scale ultrasound shows a hypoechoic mass (arrow) in the liver. (**B**) The nodule (arrow) shows homogeneous hyperenhancement at 24 s in the arterial phase of Contrast-Enhanced Ultrasound (CEUS). (**C**) Early washout is apparent (arrow) at 40 s in the portal venous phase. (**D**) The nodule (arrow) shows a contrast defect in the Kupffer phase. (**E**) Pre-contrast CT shows a subtle hypodense mass (arrow). (**F**) The mass shows ring enhancement (arrow) in the arterial phase of Contrast-Enhanced CT (CECT). (**G**,**H**) Intratumoral enhancement (arrows) gradually increases in the portal venous and equilibrium phases.

**Table 1 diagnostics-10-00828-t001:** Modified Contrast-Enhanced Ultrasound Liver Imaging Reporting and Data System (CEUS LI-RADS) criteria.

Rating Criteria with Nodule Size	Arterial Phase	Early Washout (<60 s) or Not	Kupffer Phase (≥10 min)
**LR-5**			
≥1 cm	Hyperenhancement (not rim, not peripheral discontinuous globular hyperenhancement)	No	Hypoechoic
**LR-M**			
Nodule size not considered	Rim hyperenhancement or not rim, not peripheral discontinuous globular hyperenhancement	Yes	Hypoechoic
**LR-4**			
≥2 cm	No hyperenhancement	No	Hypoechoic
≥1 cm	Not rim, not peripheral discontinuous globular hyperenhancement	No	Isoechoic or Hyperechoic
<1 cm	Not rim, not peripheral discontinuous globular hyperenhancement	No	Hypoechoic
**LR-3**			
<2 cm	No hyperenhancement	No	Any
≥2 cm	No hyperenhancement	No	Isoechoic or Hyperechoic
<1 cm	Not rim, not peripheral discontinuous globular hyperenhancement	No	Isoechoic or Hyperechoic
**LR-2**			
<1 cm	Iso-enhancement	No	Isoechoic or Hyperechoic
**LR-1**			
Nodule size not considered	Definitely benign arterial phase pattern (cyst, hemangioma, focal fatty deposition/sparing, or other definitely benign finding)	No	N/A

**Table 2 diagnostics-10-00828-t002:** Clinical and histopathologic data.

Characteristic	Result
**Median age (y) ***	70.0 (54.5–78.0)
**Sex**	
Male	74 (71.2%)
Female	30 (28.8%)
**Median nodule size (mm) ***	17.9 (13.1–28.2)
**Liver disease etiology**	
HCV	36 (34.6%)
HBV	27 (26.0%)
Alcohol	26 (25.0%)
NASH	11 (10.6%)
HCV + HBV	1 (1.0%)
AIH	1 (1.0%)
Unknown	2 (1.9%)
**Presence of cirrhosis**	80 (87.9%)
**Histopathologic analysis**	
HCC	64 (61.5%)
Well-differentiated	28
Moderately differentiated	32
Poorly differentiated	4
ICC	6 (5.8%)
Metastasis	9 (8.7%)
FNH	5 (4.8%)
Dysplastic nodule	3 (2.9%)
AML	1 (1.0%)
Focal fatty change	1 (1.0%)
Diffuse large B-cell lymphoma	1 (1.0%)
Adrenal rest tumor	1 (1.0%)
**No histopathologic analysis**	
Contrast-enhanced CT or MRI and follow-up	
Hemangioma	10 (9.6%)
FNH	2 (1.9%)
Focal spared lesion	1 (1.0%)

HCV, hepatitis C virus; HBV, hepatitis B virus; NASH, non-alcoholic steatohepatitis; AIH, autoimmune hepatitis; HCC, hepatocellular carcinoma; ICC, intrahepatic cholangiocarcinoma; FNH, focal nodular hyperplasia; AML, angiomyolipoma; y, year. * Data in parentheses are interquartile range.

**Table 3 diagnostics-10-00828-t003:** Percentages of hepatocellular carcinoma (HCC) and non-HCC malignancies in each modified CEUS LI-RADS category.

Category	No. of Nodules (% of total)	Percentage HCC	Percentage Non-HCC Malignancy	Histopathologic Analysis	Contrast-Enhanced CT or MRI and Follow-Up
LR-1	7 (6.7)	0	0	0	7
LR-2	1 (1.0)	0	0	1	0
LR-3	10 (9.6)	70.0 (7/10)	0	9	1
LR-4	16 (15.3)	37.5 (6/16)	0	13	3
LR-5	48 (46.2)	93.8 (45/48)	0	46	2
LR-M	22 (21.2)	27.3 (6/22)	68.2 (15/22)	22	0

HCC, hepatocellular carcinoma.

**Table 4 diagnostics-10-00828-t004:** Distribution of nodules in modified CEUS LI-RADS categories.

Total Nodules (104) *n* (%)	LR-1 7 (6.7)	LR-2 1 (1.0%)	LR-3 10 (9.6%)	LR-4 16 (15.3%)	LR-5 48 (46.2%)	LR-M 22 (21.2%)
HCC	0	0	7 (70.0%)	6 (37.5%)	45 (93.8%)	6 (27.3%)
Metastasis	0	0	0	0	0	9 (40.9%)
ICC	0	0	0	0	0	6 (27.3%)
Lymphoma	0	0	0	0	0	1 (4.6%)
Hemangioma	6 (85.7%)	0	0	2 (12.5%)	2 (4.2%)	0
FNH	0	0	1 (10.0%)	6 (37.5%)	0	0
DN	0	0	2 (20.0%)	1 (6.3%)	0	0
AML	0	0	0	1 (6.3%)	0	0
Adrenal rest tumor	0	0	0	0	1 (2.1%)	0
Focal fatty area	0	1 (100%)	0	0	0	0
Focal spread lesion	1 (14.3%)	0	0	0	0	0

HCC, hepatocellular carcinoma; ICC, intrahepatic cholangiocarcinoma; FNH, focal nodular hyperplasia; DN, dysplastic nodule; AML, angiomyolipoma.

**Table 5 diagnostics-10-00828-t005:** Imaging characteristics of all liver nodules.

Imaging Features	Malignant Lesions				Benign Lesions							Total (*n* = 104)
	HCC (*n* = 64)	Metastasis (*n* = 9)	ICC (*n* = 6)	Lymphoma (*n* = 1)	Hemangioma (*n* = 10)	FNH (*n* = 7)	DN (*n* = 3)	AML (*n* = 1)	Adrenal Rest Tumor (*n* = 1)	Focal Fatty Area (*n* = 1)	Focal Spared Lesion (*n* = 1)	
**Gray-scale echogenicity**												
Hyperechoic	13	3	1	0	2	3	1	0	1	1	0	25
Isoechoic	10	1	0	0	1	1	0	0	0	0	0	13
Hypoechoic	41	5	5	1	7	3	2	1	0	0	1	66
**Arterial phase**												
Hyperenhancement	56	2	5	1	7	7	1	1	1	0	0	81
Diffuse	56	2	4	1	4	7	1	1	1	0	0	77
Rim	0	0	1	0	0	0	0	0	0	0	0	1
Peripheral nodular	0	0	0	0	3	0	0	0	0	0	0	3
Iso-enhancement	7	4	0	0	0	0	1	0	0	1	1	14
Hypo-enhancement	1	3	1	0	3	0	1	0	0	0	0	9
**Kupffer phase**												
Isoechoic	11	0	0	0	7	5	3	1	0	1	1	29
Hypoechoic	53	9	6	1	3	0	0	0	1	0	0	73
Marked	16	9	6	1	0	0	0	0	0	0	0	32
Mild	37	0	0	0	3	0	0	0	1	0	0	41
Hyperechoic	0	0	0	0	0	2	0	0	0	0	0	2
**Washout**												
<60 s	6	9	6	1	0	0	0	0	0	0	0	22

HCC, hepatocellular carcinoma; ICC, intrahepatic cholangiocarcinoma; FNH, focal nodular hyperplasia; DN, dysplastic nodule; AML, angiomyolipoma.

**Table 6 diagnostics-10-00828-t006:** Diagnostic performance of the modified CEUS LI-RADS LR-5 and LR-M categories.

Criteria	Se (%)	95% CI	Sp (%)	95% CI	PPV (%)	95% CI	NPV (%)	95% CI	Accuracy (%)	95% CI
LR-5	70.3	57.6–81.1	92.5	79.6–98.4	93.8	82.8–98.7	66.1	52.2–78.2	78.7	69.7–86.2
LR-M	68.2	45.1–86.1	100	93.5–100	100	69.8–100	92.1	84.5–96.8	93.3	86.6–97.3

Se, sensitivity; CI, confidence interval; Sp, specificity; PPV, positive predictive value; NPV, negative predictive value.

**Table 7 diagnostics-10-00828-t007:** Relationship between modified CEUS LI-RADS classification and HCC histologic differentiation.

CEUS LI-RADS	Well-Differentiated	Moderately Differentiated	Poorly Differentiated
LR-3	6	1	0
LR-4	5	1	0
LR-5	17	28	0
LR-M	0	2	4
Total	28	32	4

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
