# Peer review of "Usefulness of Modified CEUS LI-RADS for the Diagnosis of Hepatocellular Carcinoma Using Sonazoid"

_diagnostics, 2020, doi:10.3390/diagnostics10100828_

Round 1

Reviewer 1 Report

I read with pleasure this paper about a proposed modified version of CEUS LI-RADS specifically designed for Sonazoid. Indeed, data about the clinical use of Sonazoid are limited to Japan and Korea. In these terms, this work is interesting: the idea proposed by the Authors is very simple (substitute the type of wash-out in the late phase with the findings in the vascular phase) yet potentially effective. As a matter of fact, such proposal would have required a multicenter prospective study with a validation cohort to obtain strong evidence. With this important limitation in mind, this paper still represents an interesting proof of concept for future studies and is worth of attention. I have some specific comments:

MAJOR POINTS

  1. The Authors state that they enrolled patients at risk of HCC. Whil this vague definition is endorsed by the CEUS LI-RADS, the actual applicability of these criteria should be strictly limitied to cirrhotic patients. The peculiar vascular hemodynamics of HCC can be appreciated only in the setting of a cirrhotic liver. As a critical point, the Authors should clearly specify how many patients of the total population (n=104) are actually cirrhotics and which methods were used to diagnose liver cirrhosis (clinical findings, laboratory alterations, conventional US findings, elastosonography, biopsies, a combination of all these elements?).
  2. The choice to adopt other imaging techniques as reference standards is debatable at least. I understand that, given the nature if this study, it was impossible to propose a biopsy for allegedly benign lesions. Moreover, the number of biopsies is acceptable, so this is not a “fatal” flaw but still needs a more accurate discussion in the study limitations.
  3. Interestingly, the Authors found that 7/10 LR3 nodules are HCC. This proportion is even higher than that of LR-4 nodules. According of the LI-RADS classification, LR-3 are nodules at relatively low risk of malignancy. The Authors should be encouraged to report which specific characteristic of these 7 nodules were “misleading”. For instance, were they classified as LR-3 because of dimension<20 mm, absence of arterial hyperenhancement, or regular post-vascular phase? I suggedto dedicate either a small pragrpah or a Table to this specific question.

MINOR POINTS

  1. Figure 1: a ≈symbol might be more appropriate than a ≤  symbol when defining the timing of the post-vascular phase.
  2. Table 2 and Results: when describing the distribution of nodule dimension, median and IQR could be more informative than mean and SD (please check that his variable was normally distributed).

Author Response

Responses to the Editor and Reviewers

To the Editors and Reviewers:

Thank you very much for your helpful and constructive comments on our paper (diagnostics-928756) entitled "Usefulness of Modified CEUS LI-RADS for the Diagnosis of Hepatocellular Carcinoma Using Sonazoid". We have revised our manuscript based on all of the reviewers' comments.

Please see the annotated revised manuscript, which was made changes to address the reviewers’ comments. Also, our detailed responses to each of your comments are given below.

Reviewer 1

Comments and Suggestions for Authors

I read with pleasure this paper about a proposed modified version of CEUS LI-RADS specifically designed for Sonazoid. Indeed, data about the clinical use of Sonazoid are limited to Japan and Korea. In these terms, this work is interesting: the idea proposed by the Authors is very simple (substitute the type of wash-out in the late phase with the findings in the post vascular phase) yet potentially effective. As a matter of fact, such proposal would have required a multicenter prospective study with a validation cohort to obtain strong evidence. With this important limitation in mind, this paper still represents an interesting proof of concept for future studies and is worth of attention. I have some specific comments:

Reply: Thank you for your instructive comments.

MAJOR POINTS

  1. The Authors state that they enrolled patients at risk of HCC. While this vague definition is endorsed by the CEUS LI-RADS, the actual applicability of these criteria should be strictly limited to cirrhotic patients. The peculiar vascular hemodynamics of HCC can be appreciated only in the setting of a cirrhotic liver. As a critical point, the Authors should clearly specify how many patients of the total population (n=104) are actually cirrhotic and which methods were used to diagnose liver cirrhosis (clinical findings, laboratory alterations, conventional US findings, elastography, biopsies, a combination of all these elements?).

Reply: Thank you for your suggestion. Based on our pathologic database, among 91 patients who undertook liver histological examination, 80 patients (87.9%) had liver cirrhosis. We added these descriptions in “3.1. Patient Characteristics” and “Table 2”.

  1. The choice to adopt other imaging techniques as reference standards is debatable at least. I understand that, given the nature if this study, it was impossible to propose a biopsy for allegedly benign lesions. Moreover, the number of biopsies is acceptable, so this is not a “fatal” flaw but still needs a more accurate discussion in the study limitations.

Reply: Thank you for your important suggestion. We added one sentence to the “Discussion” section (page 11, first paragraph).

  1. Interestingly, the Authors found that 7/10 LR3 nodules are HCC. This proportion is even higher than that of LR-4 nodules. According of the LI-RADS classification, LR-3 are nodules at relatively low risk of malignancy. The Authors should be encouraged to report which specific characteristic of these 7 nodules were “misleading”. For instance, were they classified as LR-3 because of dimension<20 mm, absence of arterial hyperenhancement, or regular post-vascular phase? I suggested to dedicate either a small paragraph or a Table to this specific question.

Reply: Thank you for your important suggestions. As you suggested, we added some descriptions on LR-3 and LR-4 in 3.2. (page 6) and Table 4, which was newly established.

MINOR POINTS

  1. Figure 1: a ≈symbol might be more appropriate than a ≤ symbol when defining the timing of the post-vascular phase.

Reply: Thank you for your suggestion. We corrected it following your advice.

  1. Table 2 and Results: when describing the distribution of nodule dimension, median and IQR could be more informative than mean and SD (please check that his variable was normally distributed).

Reply: Thank you for your suggestion. We corrected it following your advice.

Reviewer 2 Report

This study is useful for classifying suspected HCCs with Sonazoid. The subject matter of this work is laudable and of interest to radiologist and hepatologist. From the reviewer point of view, there are some issues in the results that should be made clear. In these results, LR-4 including the HCCs about 40%. I’d like to know the details about the pathological diagnosis of LR-4 nodules (7 lesions are not HCCs). In table 4, each pathological diagnosis should be classified by LR categories (except for LR-1). In table 5, please show the details about LR-4 categories.

Minor comment

Page 8

 Table 6 is not in the paper. I can’t check the table.

Author Response

Responses to the Editor and Reviewers

Reviewer 2

Comments and Suggestions for Authors

This study is useful for classifying suspected HCCs with Sonazoid. The subject matter of this work is laudable and of interest to radiologist and hepatologist. From the reviewer point of view, there are some issues in the results that should be made clear. In these results, LR-4 including the HCCs about 40%. I’d like to know the details about the pathological diagnosis of LR-4 nodules (7 lesions are not HCCs). In table 4, each pathological diagnosis should be classified by LR categories (except for LR-1). In table 5, please show the details about LR-4 categories.

Reply: Thank you for your important suggestions. As you suggested, we added some descriptions on LR-3 and LR-4 in 3.2. (page 6) and Table 4, which was newly established.

Minor comment

Page 8

 Table 6 is not in the paper. I can’t check the table.

Reply: Sorry for confusing. We added the Table corresponding to Table 7.

Reviewer 3 Report

It is an article to prove the usefulness of modified CEUS LI-RADS in the diagnosis of HCC and non-HCC malignant liver tumors. Authors introduced the criteria of modified CEUS LI-RADS, collected some cases checked by CEUS LI-RADS, and compared the diagnosis with their corresponding histopathologic examination or dynamic CT and MRI with no change in size over at least 1-year follow-up period. The sensitivity, specificity, PPV and NPV of modified CEUS LI-RADS had been assessed. At last, authors found LR-5 and LR-M of CEUS LI-RADS were predictors of HCC and non-HCC malignant liver tumors. It is an interesting article, which would help readers understand the value of modified CEUS LI-RADS in HCC diagnosis. The experiment designs are good, and data is credible, but there is a problem in formulation.

  1. Page 3, 2.3. Reference Standard “The clinical and histopathologic data are shown in Table 1.” but there is not clinical data, please change to “the criteria of modified CEUS LI-RADS are shown in Table 1”
  2. The accuracy of CT and MRI is better than ultrasonic image, maybe authors need to add a few sentences to emphasize the value of ultrasonic image, maybe checking time is shorter, cheaper?

Author Response

Responses to the Editor and Reviewers

Reviewer 3

Comments and Suggestions for Authors

It is an article to prove the usefulness of modified CEUS LI-RADS in the diagnosis of HCC and non-HCC malignant liver tumors. Authors introduced the criteria of modified CEUS LI-RADS, collected some cases checked by CEUS LI-RADS, and compared the diagnosis with their corresponding histopathologic examination or dynamic CT and MRI with no change in size over at least 1-year follow-up period. The sensitivity, specificity, PPV and NPV of modified CEUS LI-RADS had been assessed. At last, authors found LR-5 and LR-M of CEUS LI-RADS were predictors of HCC and non-HCC malignant liver tumors. It is an interesting article, which would help readers understand the value of modified CEUS LI-RADS in HCC diagnosis. The experiment designs are good, and data is credible, but there is a problem in formulation.

  1. Page 3, 2.3. Reference Standard “The clinical and histopathologic data are shown in Table 1.” but there is not clinical data, please change to “the criteria of modified CEUS LI-RADS are shown in Table 1”

Reply: Thank you for your suggestion. We corrected it following your advice.

  1. The accuracy of CT and MRI is better than ultrasonic image, maybe authors need to add a few sentences to emphasize the value of ultrasonic image, maybe checking time is shorter, cheaper?

Reply: Thank you for your suggestion. However, we did not make any comparison between CT/MRI and US. The description may not be relevant to this main topic in the article.
